# Utility of Metagenomic Next-Generation Sequencing in Infective Endocarditis: A Systematic Review

**DOI:** 10.3390/antibiotics11121798

**Published:** 2022-12-11

**Authors:** Sara F. Haddad, Daniel C. DeSimone, Supavit Chesdachai, Danielle J. Gerberi, Larry M. Baddour

**Affiliations:** 1Division of Public Health, Infectious Diseases and Occupational Medicine, Department of Medicine, Mayo Clinic College of Medicine and Science, Mayo Clinic, Rochester, MN 55905, USA; 2Department of Cardiovascular Disease, Mayo Clinic College of Medicine and Science, Mayo Clinic, Rochester, MN 55905, USA; 3Mayo Clinic Library Services, Mayo Clinic College of Medicine and Science, Mayo Clinic, Rochester, MN 55905, USA

**Keywords:** infective endocarditis, cardiovascular implantable electronic devices, device-related infections, metagenomic next-generation sequencing, diagnosis, molecular

## Abstract

Blood cultures have been the gold standard for identifying pathogens in infective endocarditis (IE). Blood culture-negative endocarditis (BCNE), however, occurs in 40% or more of IE cases with the bulk of them due to recent antibiotic exposure prior to obtaining blood cultures. Increasingly, molecular techniques are being used for pathogen identification in cases of BCNE and more recently has included metagenomic next-generation sequencing (mNGS). We therefore performed a literature search on August 31, 2022, that assessed the mNGS in IE and 13 publications were identified and included in a systematic review. Eight (61.5%) of them focused only on IE with mNGS performed on cardiac valve tissue in four studies, plasma in three studies and cardiac implantable electronic devices (CIED) in one study. Gram-positive cocci, including *Staphylococcus aureus* (n = 31, 8.9%), coagulase-negative staphylococci (*n* = 61, 17.6%), streptococci (*n* = 130, 37.5%), and *Enterococcus faecalis* (*n* = 23, 6.6%) were the predominant organisms identified by mNGS. Subsequent investigations are needed to further define the utility of mNGS in BCNE and its impact on patient outcomes. Despite some pitfalls, mNGS seems to be of value in pathogen identification in IE cases, particularly in those with BCNE. This study was registered and on the Open Science Framework platform.

## 1. Introduction

Traditional microbiologic laboratory tools such as specimen cultures and serology are often sufficient for the identification of most pathogens [1]. However, the availability of laboratory results may be delayed and result in prolonged empiric antimicrobial coverage which may be inadequate and result in poor patient outcomes, increased healthcare costs, more drug adverse events, and selection for antimicrobial resistance [2].

Infective endocarditis (IE) is a life-threatening syndrome that can be impacted by limitations of these laboratory tools in the identification of a pathogen. In particular, the prevalence of “blood culture-negative endocarditis” (BCNE) has been as high as 40 to 71% in recent studies [3,4]. Recent antibiotic exposure has been responsible for the large majority of BCNE cases. Fortunately, advances in molecular diagnostic techniques have resulted in an enhancement in pathogen identification in cases of BCNE. The use of 16S rRNA PCR/sequencing of cardiac valve tissue, in particular, has gained popularity in pathogen identification in cases of BCNE and has been reported to be associated with improved patient outcomes [5,6]. However, there are limitations of the 16S rRNA methodology which can include cross-contamination of tissues and identification of non-viable organisms from prior infections due to the persistence of microbial deoxyribonucleic acid, and the necessity of assuming the type of microbes in the sample and using the adequate primer [7].

Metagenomic next-generation sequencing (mNGS) has been recently proposed as a novel diagnostic platform in pathogen identification in cases of BCNE. In contrast to traditional Sanger sequencing which determines a sample’s sequence one section at a time, mNGS is a high-throughput technique encompassing genome-wide areas [8]. mNGS uses parallel and simultaneous sequencing of multiple gene fragments to determine the sequence of nucleic acid content from a sample. Bioinformatic analyses then match sequences to a bank of reference genomes that can identify bacteria, viruses, fungi, and parasites [9]. This improves sequencing time while increasing coverage quality and data output [10,11].

Analysis of resected native and prosthetic valves by mNGS was first described in a case series in 2017 [12] with numerous subsequent reports of its utility in BCNE cases. Because mNGS is being performed in an increasing number of clinical laboratories, this systematic review was conducted to provide a contemporary evaluation of mNGS for IE, notably in cases of BCNE.

## 2. Methods

The literature was searched by a medical librarian (D.J.G) for the concepts of mNGS and IE or cardiovascular implantable electronic device (CIED) infections. Search strategies were created using a combination of keywords and standardized index terms. Searches were run on 31 August 2022, in Ovid Cochrane Central Register of Controlled Trials, Ovid Embase, Ovid Medline, Scopus, and Web of Science Core Collection. All results were exported to EndNote X9.3.1 (Clarivate, Philadelphia, PA, USA) and deduplication was performed in Covidence systematic review software (Veritas Health Innovation, Melbourne, Australia). Key words included: infective endocarditis, cardiac implantable electronic devices, and high-throughput or metagenomic next-generation sequencing, which were selected by OR & AND operators in combination. Full search strategies are provided in the appendix. A screening based on titles and abstracts was done by two investigators (L.M.B. and S.F.H.) independently followed by a review of all potentially relevant papers. Any conflict between the first two researchers was resolved by study review and agreement. The Preferred Reporting Items for Systematic Reviews and Meta-Analysis (PRISMA) guideline was followed for this review, which is a well-recognized guideline for systematic reviews.

Study criteria for inclusion were: (1) prospective/retrospective cohort, case–control, cross-sectional studies; (2) adults and pediatric patients; (3) native, prosthetic or device-related endocarditis; (4) all studies that examined the diagnostic value of mNGS for IE and CIED value (reporting at least one of the following: sensitivity, specificity, positive predictive values (PPV), or negative predictive values (NPV)); (5) studies suggesting ways to enhance the mNGS in IE; (6) studies that included mNGS for patients with IE, but not limited to IE. Case reports, case series editorials, conference abstracts and animal studies were excluded, as well as studies that used solely different diagnostic tools for pathogen identification without mNGS. Results were limited to the English language. Data including the first author’s last name, the year of publication, the country, the study period, the types of study, the sensitivity, and the specificity of mNGS were retrieved and recorded for each study. The current systematic review was registered on the Open Science Framework (OSF) platform and received approval on September 7, 2022 (https://doi.org/10.17605/OSF.IO/EKSYX, accessed on 9 December 2022).

## 3. Results

Searches were run on 31 August 2022, without time constraints in Ovid Cochrane Central Register of Controlled Trials (1991+), Ovid Embase (1974+), Ovid Medline (1946+ including epub ahead of print, in-process and other non-indexed citations), Scopus (1788+), and Web of Science Core Collection (Science Citation Index Expanded 1975+ and Emerging Sources Citation Index 2015+).

A total of 4506 citations were retrieved (Figure 1). Deduplication was performed in Covidence leaving 2791 citations. Abstracts of the remaining papers were then examined, and 2647 irrelevant articles were eliminated. Following that, the full text of the remaining 144 articles was evaluated, resulting in the deletion of 131 more articles due to irrelevance, study design not matching with inclusion criteria, studies limited to preliminary data where the abstract was only available, studies limited to laboratory study only or a different diagnostic tool was used for IE diagnosis. Ultimately, 13 publications were included in the systematic review; eight studies had more details that were evaluated in this review. The remaining five publications had scant information but were also included.

Six studies were conducted in the USA, three in China, two in France, one in Spain, and one in Denmark. Nine studies were retrospective cohorts and four were prospective cohorts. It is to be noted that the study by To RK and colleagues was identified as case series by the authors; however, they performed a retrospective chart review from January 2017 to January 2020 and included all subjects of 0–21 years of age who had a diagnosis of definite IE and plasma mNGS, thus qualifying for a retrospective cohort. Eight of the studies included 373 patients and evaluated mNGS as a novel diagnostic tool in IE pathogen identification in comparison to other tools that have been available for longer periods of time, including 16S PCR, serology, and culture-based methods. In four of the eight studies, mNGS was performed using cardiac valve tissue and compared results to those of cultures of blood and valve tissue. In three investigations, mNGS was performed using plasma, with one study comparing the results between whole blood and plasma. Only one study explored the use of mNGS and sonication in device-related infections (Table 1). Of note, a 2018 study from Cheng J and colleagues [13] assessed the feasibility of a metagenomic approach to detect the causative pathogens in resected valves from IE patients; however, this work was not included in our systematic review as the data in this earlier investigation were included in the subsequent report that involved a longer study period with a larger number of cases [14].

As expected, Gram-positive cocci, including *Staphylococcus aureus* (*n* = 31, 8.9%), coagulase-negative staphylococci (*n* = 61, 17.6%), streptococci (*n* = 130, 37.5%), and *Enterococcus faecalis* (*n* = 23, 6.6%) were the predominant organisms identified by mNGS. Other less common causes of IE identified are listed in Table 1. Of note, recent antibiotic exposure was reported in 43.4% to 100% of cases.

The average turnaround time, which represents the time between sample collection to the time of availability of test results, was 3.2 days (SD 1.2 days) for mNGS versus 11 days for 16S PCR in a study [19], and 48 hours versus 1 week for culture method in another study [16].

Sensitivity and specificity were collected for the studies and differed between cardiac valve mNGS and plasma mNGS, with a wide range varying between patients who had positive and negative blood or valve cultures. Culture-positive results in IE patients were set as the gold standard to assess the sensitivity and specificity of mNGS in the etiological diagnosis of IE. The sensitivity ranged between 85.9 and 100% for cardiac valve mNGS, and between 47% and 80% for plasma mNGS, with a specificity of 72.7% to 100% in patients with cardiac valve mNGS, and a specificity of 71.4% to 100% in patients with plasma mNGS. The PPV of mNGS was only reported in two studies, with a value around 97%; however, the NPV differed between these two studies (85.7 and 36.4%) [14,16]. Sensitivity and PPV of generators and lead tip sonication and NGS were lower for systemic device-related infections (DRI) (Sen: 18%, PPV ~ 50%; Sen: 48%, PPV ~ 90%, respectively) compared to those of blood culture (Sen: 93%, PPV ~ 99%) [21]. Table 2 outlines the sensitivity and specificity of mNGS compared to that of more established diagnostic testing, with the positive and negative predictive values.

As for the five remaining studies, they included only scant clinical information related to the definition of IE, a non-IE focused study, and lacked comparison to the results of conventional IE pathogen identification methods. Three retrospective cohort studies on plasma cell-free DNA-based pathogen identification had a few patients with IE among other infectious syndromes [22,23,24]. Million M et al. and Bouchiat et al. proposed two different steps to enhance the results of mNGS, microdissection combined with human DNA depletion [25] and a nanopore-based 16S rRNA metagenomic approach [26].

## 4. Discussion

This is the first systematic review to investigate mNGS in the identification of IE pathogens. We included 13 studies with eight of them evaluating the performance of mNGS for IE with a total of 373 patients. The five remaining studies did not provide the sensitivity, specificity, PPV or NPV of mNGS but were also included. The sensitivity and specificity in cardiac valve and plasma mNGS differed among the studies and patients’ groups with higher sensitivities for cardiac valve mNGS (85.9 to 100%) as compared to that for plasma mNGS (47% to 80%). Specificity was higher in patients who also had either positive blood or valve cultures (100%) compared to those with negative cultures (71.4%). Although blood cultures have long been the gold standard for pathogen identification in IE, novel diagnostic methods such as mNGS may be more sensitive (see Table 2). For example, mNGS with blood or valve cultures improved diagnostic accuracy in one study (Sen 89.9%, Sp 72.7%, AUC 0.813) [16]. Moreover, although rare, cases due to polymicrobial infection were identified using mNGS, but not in culture-based methods, indicating mNGS may have better performance in identifying polymicrobial infections [16,17].

The goal of pathogen identification is to ensure that the early administration of optimal antimicrobial therapy is initiated to hopefully improve patient outcomes. There are several laboratory tools available to achieve this (Figure 2) with pros and cons identified for each methodology (Table 3). At present, blood cultures remain key in pathogen identification; thus, the remaining laboratory diagnostic options are largely relegated to a secondary position in IE. Because IE is a life-threatening infection, a prompt pathogen diagnosis is critical. Because clinical laboratories vary in availability of diagnostic techniques, immediate discussion with laboratory personnel is key when cases of BCNE are considered. Sequencing of subsequent non-culture-based screening will be based on this discussion; thus, “positioning” of specific tools in an algorithm is difficult for a “one size fits all” scenario. While a variety of laboratory techniques are included in Figure 2 and Table 3, it is beyond the scope of the manuscript to review aspects of each laboratory tool mentioned.

However, it remains essential to correlate the mNGS to the clinical context of the patient as there may be false positives as suggested in some studies possibly due to laboratory contamination or detection of normal skin flora [15,16,23]. Sequencing results include many parameters that reflect the number of distinct and unique sequences aligned to a microorganism’s genome. These parameters are related to the number of microorganisms in a specimen, the size of the microorganism genome, the length of nucleic acid sequence present in the microorganism’s whole genome and the amount extracted from a specimen. Higher value refers to the higher credibility of the existence of a pathogen. In the study of Cai S et al., the parameters of all IE valve mNGS included: stringent mapped reads number (SMRN) 10100, stringent mapped reads number of genera (SMRNG) 12072, a relative abundance of genera (RAG) of 95.5%, relative abundance of species (RAS) of 70.1%, and a coverage rate (CV) of 26.6%. However, these parameters were significantly higher in culture-positive IE cases than in those with culture-negative IE (*p* < 0.05) [15]. To consider mNGS positive, there is a certain consensus on the threshold (90%), above which a pathogen is considered causative of infection [17,18]. This threshold should be interpreted carefully as lower count reads can also be clinically relevant. In fact, bacterial pathogenicity is dependent on both endogenous and exogenous factors and is not limited to bacterial counts alone [27]; thus, the high sensitivity of mNGS in the detection of even minute traces of bacterial DNA, regardless of the organism’s replication performance, may be useful.

The narrow spectrum of specific 16s rRNA sequencing assays limits its usage in clinical practice as it can only amplify genes from certain bacteria only with targeted and predetermined primers or probes. mNGS offers a wider range of microorganisms’ detection, including viruses and fungi [20,24,27]. Pathogens that are unexpected, infrequent, or new may avoid identification by serological testing and 16S rRNA PCR. Moreover, the potential of mNGS to detect antimicrobial resistance (AMR) genes could greatly impact antimicrobial management strategies. Cheng et al. for example, identified a mecA gene with high abundance among nine structured antibiotic resistance genes using NGS findings of a methicillin-resistant *Staphylococcus aureus* (MRSA) strain already validated by a VITEK 2 COMPACT antibiotic susceptibility analyzer and Sanger sequencing. Future studies should focus on advances in AMR gene analysis by mNGS, as the detection of an AMR gene does not necessarily translate to the phenotypic expression of in vitro drug resistance, either because of poor expression, silencing or inactivation of the respective gene [14].

The average turnaround time in availability of test results is key in the diagnosis and management of patients for many scenarios related to either infectious and non-infectious conditions; a quicker turnaround time should accelerate the institution of appropriate medical or surgical interventions. Indeed, mNGS had an overall shortened turnaround time as compared to that for 16S rRNA PCR among studies included in the current systematic review. For example, the average turnaround time was 11 days for 16S rRNA PCR results from a reference laboratory compared to 3.2 days (SD 1.2 days) for mNGS results [19]. Similarly, most mNGS results were available within 48 hours in the study by Zeng and colleagues [16].

mNGS can also detect non-viable or unculturable bacteria provided that microbial nucleic acids are present, overcoming the limited sensitivity in traditional culture-based approaches due to prior antimicrobial administration. In fact, the duration of positive results of microbial cell-free DNA (mcfDNA) may be clinically useful in pathogen identification and deserves highlighting. The median length of positivity from the start of antibiotic therapy was approximately 38.1 days for mcfDNA versus 3.7 days for a positive blood culture result (proportional odds, 2.95; *p* = 0.03) using a semi-parametric interval censored survival analysis [20]. Although further study is needed to evaluate the utility of mcfDNA, it is conceivable that it could be useful in pathogen identification among BCNE cases, particularly when there is a delay in IE diagnosis. For example, of nine BCNE cases, mNGS identified one patient with *Bartonella quintana* and eight patients with *Coxiella burnetii* [16]. A prolonged duration of positivity, however, may be confusing as bacterial nucleic acid debris can remain in tissue without being clinically pertinent [28,29,30].

Some clinical factors may be associated with the pathogen detection rate of mNGS. Pre-operative peripheral white blood cell (WBC) count and neutrophils have been identified as independent factors affecting the detection rate of mNGS with *p* values of 0.029 and 0.046, respectively [16]. Cai S et al. also suggested that in patients with an acute course who have high fever, elevated peripheral WBC count, and increased serum inflammatory markers, the sensitivity of blood mNGS may be enhanced in pathogen identification compared to that among patients with a more indolent subacute course of IE [15]. Other factors affecting the detection rate by mNGS are blood culture positivity and cycle threshold (CT) values at the time of specimen collection [18]. In fact, this study showed that having a positive blood culture at the time of targeted metagenomic sequencing (tMGS) specimen collection increased the likelihood of a positive result from whole blood, plasma, or both (*p* < 0.05), and samples with CT values < 37 increased the likelihood of a positive tMGS from plasma (*p* = 0.04), but not from whole blood or both combined (*p* > 0.05) [18].

Because mNGS includes quantification in molecules per milliliter, this may be useful in both assessing the significance of an identified organism, and a response to antimicrobial therapy with a drop in quantification over time that would predict a corresponding decrease in microbiologic load and has been demonstrated by serial mNGS screening [19]. Additional investigations, however, are needed to further substantiate the utility of mNGS in predicting patient response to treatment.

Ultimately, the primary goal of mNGS is to ensure that appropriate antimicrobial therapy is being administered in patients with BCNE with the expectations that outcomes will be improved. To RK and colleagues, for example, reported that four of eight IE patients underwent revision of antimicrobial regimens, based on mNGS results, to enhance the likelihood of treatment success [19]. This was also the case in the study of Cai S et al., where antibiotic therapy was de-escalated to cephalosporin and aminoglycosides for six culture-negative IE subjects following pathogen identification by valve mNGS screening of valve tissue [15]. In addition, mNGS may be also important in enhancing antimicrobial stewardship, and reducing both adverse drug events and selection for antimicrobial resistance.

Cardiac device-related IE deserves highlighting. Olsen and colleagues focused on CIED infections and described results of the only investigation to date to use mNGS in pathogen diagnosis [21]. Based on the blood culture-based case definition of CIED-related IE, it is not surprising that >90% of patients had positive blood cultures. Remarkably, less than half of the extracted leads and less than one-fifth of the generators had matching pathogens for systemic DRI. These findings prompt the question whether patients with a diagnosis of CIED-related IE had device infection and whether device removal was needed. Moreover, one can question if there is a role for mNGS in blood culture-positive cases. Thus, further investigation in the role of mNGS in CIED-related IE is warranted.

As for the remaining studies, three retrospective cohort studies evaluated the clinical relevance of plasma cell-free DNA-based pathogen identification and its impact on antimicrobial management. The first was performed for multiple infection syndromes and showed that plasma mNGS can impact patient management; however, only 8.5% (*n* = 7) of the patients had a suspicion of IE [22]. In the second, mNGS was used in 16.3% (*n* = 13) of IE patients, and led to confirmation of diagnosis in two patients, de-escalation of antimicrobials in four patients, and procedure avoidance in one patient with IE [24]; and in the third only four patients with BCNE were evaluated [23], thus results were difficult to construe.

Million M et al. showed that microdissection combined with human DNA depletion was a key approach for controlling background DNA in metagenomics shotgun sequencing for pathogen identification [25]. Bouchiat et al. evaluated a nanopore-based 16S rRNA metagenomic approach—a technology that enables rapid and user-friendly library preparation, real-time data acquisition and analysis through an online platform with dedicated pipelines—and showed that setting a threshold of 1% of total reads overcame background noise and eased the interpretation of clinical samples. However, only two samples were cardiac valves; thus, it is difficult to draw conclusions regarding the applicability of this technique in IE [26].

Due to the scarcity of available investigations, we widened the scope of article selection in this systematic review which explains the heterogeneity in the study objectives included and the inconsistency in established criteria for ordering mNGS across the studies. These are important limitations as they weaken the evidence presented in the review. However, the high positive rate of mNGS compared to blood culture results supports the potential value of mNGS. Although it is tempting to speculate whether mNGS will replace 16S rRNA if the former becomes cheaper and more readily available in clinical laboratories, additional study is needed to further define mNGS in IE pathogen identification.

## 5. Conclusions

Advances in molecular techniques, including mNGS, are likely to play a pivotal role in pathogen identification and improve outcomes among patients with BCNE. Expansion of availability of mNGS from only referral to onsite laboratories continues. Additional investigations are needed to further define the proficiency of mNGS and its ultimate impact on outcomes in BCNE patients.

## Figures and Tables

**Figure 1 antibiotics-11-01798-f001:**
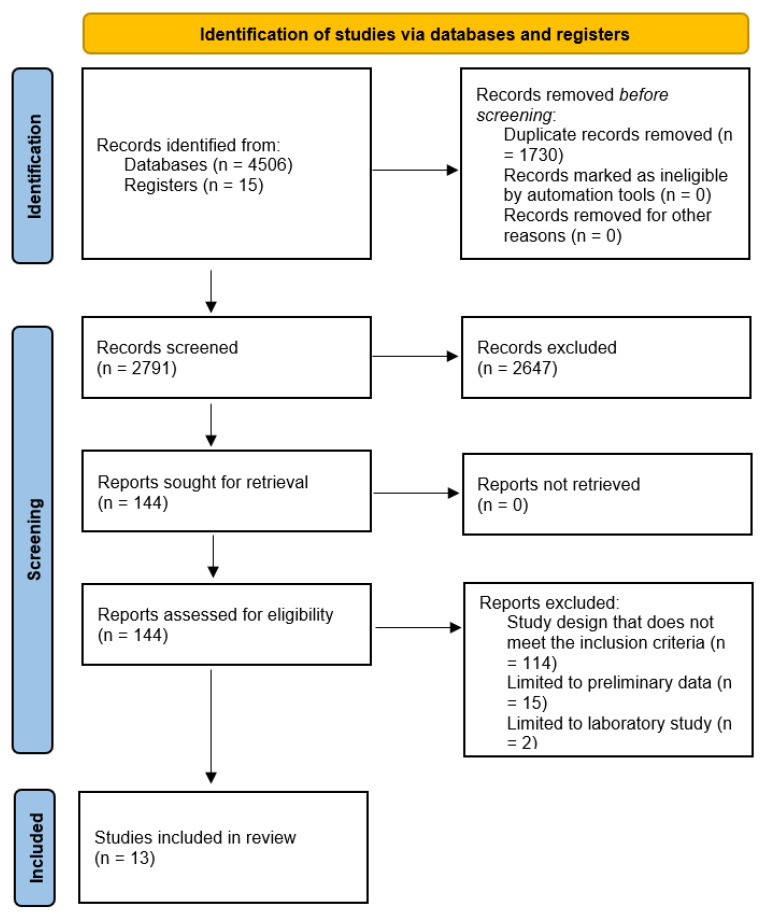
Search strategy flowchart.

**Figure 2 antibiotics-11-01798-f002:**
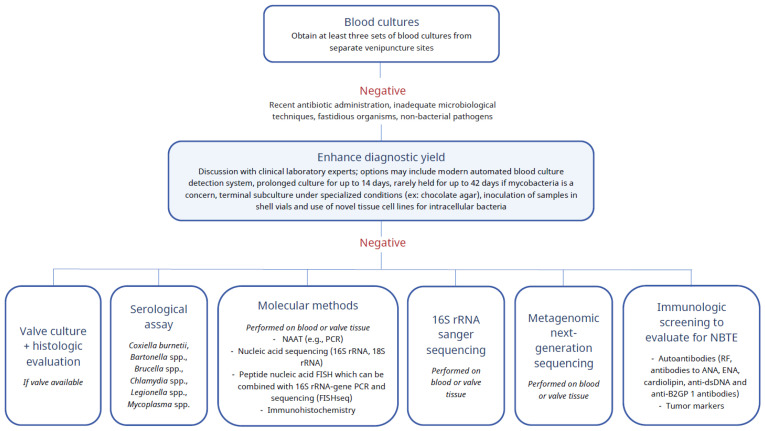
A proposed multimodal laboratory strategy for pathogen identification in patients with blood culture-negative endocarditis. anti-B2GP1: anti-beta-2-glycoproteins; anti-dsDNA: anti-double-stranded DNA; ANA: anti-nuclear antibodies; ENA: extractable nuclear antigens; FISH: fluorescence in situ hybridization; NAAT: Nucleic acid amplification test; NBTE: Nonbacterial thrombotic endocarditis; PCR: polymerase chain reaction; RF: rheumatoid factor; 16S rRNA: 16S ribosomal RNA.

**Table 1 antibiotics-11-01798-t001:** The characteristics of the studies and identified microorganisms.

Type of Specimens	Authors, Date of Publication	- Country - Study Period - Type of Study	*n* Value	Previous Antibiotic Exposure	Microbiology (*n* Values)
mNGS on **valve**	Cheng J et al. 2019 [14]	- China - April 2017–September 2018 - Retrospective cohort	**51**: 44 IE - 41 definite IE - 3 possible IE 7 negative controls	35 (79.5%)	*Staphylococcus aureus* (2) *Streptococcus* spp. (28) *Enterococcus faecalis* (1) *Abiotrophia defectiva* (2) *Pseudomonas aeruginosa* (1) *Haemophilus parainfluenzae* (1) *Bartonella quintana* (1) *Coxiella burnetii* (6)
	Cai S et al. 2021 [15]	- China - June 2018–November 2020 - Retrospective cohort	**57**: 49 IE - 28 culture positive - 21 culture negative 8 negative controls (43 NVE, 6 PVE)	46 (93.9%)	*Staphylococcus* spp. (10) *Streptococcus* spp. (26) *Enterococcus faecalis* (1) *Haemophilus parainfluenzae* (1) *Abiotrophia defectiva* (4) *Granulicatella adiacens* (2) *Gemella haemolysans* (1) *Erysipelothrix rhusiopathiae* (1) *Cardiobacterium hominis* (1) *Aggregatibacter segnis* (1) *Candida parapsilosis* (1)
	Zeng X et al. 2022 [16]	- China - May 2019–December 2020 - Prospective cohort	**110**: 99 IE 11 negative controls	43 (43.4%)	*Staphylococcus aureus* (5) *Staphylococcus* spp. (5) *Streptococcus* spp. (51) *Enterococcus faecalis* (3) *Granulicatella adiacens* (3) *Coxiella burnetii* (8) *Legionella drancourtii* (4) *Bartonella Quintana* (1) Other Polymicrobial (10 patients)
16S rRNA gene targeted NGS on heart **valve**	Santibáñez P et al. 2021 [17]	- Spain - 2009–2017 - Retrospective cohort	**27**: - 23 definite IE - 2 possible IE - 2 unavailable 4 BCNE (18 NV, 4 PV, 5 DRI)	25 (92.6%)	*Staphylococcus aureus* (1) *Staphylococcus* spp. (3) *Streptococcus* spp. (7) *Enterococcus faecalis* (4) *Haemophilus parainfluenzae* (1) *Coxiella burnetii* (1) Polymicrobial (10 patients)
**Whole Blood** and **Plasma** NGS on an Illumina MiSeqTM platform	Flurin L et al. 2022 [18]	- USA - October 2020–July 2021 - Prospective cohort, pilot study	**35**: - 28 definite IE - 7 possible IE 6 BCNE ( 13 NV, 22 PV)		*Staphylococcus aureus* (8) *Staphylococcus* spp. (3) *Streptococcus* spp. (6) *Enterococcus faecalis* (2) *Kingella* sp. (1) *Cutibacterium acnes* (1) *Corynebacterium* spp. (2) *Bartonella* sp. (1)
**Plasma** mcfDNA (Karius)	To RK et al. 2021 [19]	- USA - January 2017–January 2020 - Retrospective cohort	**10**		*Staphylococcus aureus* (1) *Staphylococcus epidermidis* (1) *Streptococcus* spp. (2) *Kingella kingae* (1) *Corynebacterium diphtheriae* (1) *Pseudomonas aeruginosa* (1) *Gemella bergeri* (1)
	Eichenberger EM et al. 2022 [20]	- USA - July 2016 -January 2018 - Prospective observational cohort study	**23**: - 23 definite IE (15 NV, 5 PV, 7 infected PM or cardioverter defibrillator lead)		*Staphylococcus aureus* (14) *Staphylococcus epidermidis* (2) *Streptococcus agalactiae* (1) *Enterococcus faecalis* (2) *Candida albicans* (2) *Pantoea ananatis* (1)
**NGS on DRI**	Olsen T et al. 2022 [21]	- Denmark - October 2016 and January 2019 - Descriptive, prospective, multicenter study	**60**: Pockets, generators, leads 41 PM, 14 ICD, 3 CRT-P, 2 CRT-D		*Staphylococcus aureus* (25) *Staphylococcus* spp. (9) *Streptococcus* spp. (9) *Enterococcus faecalis* (10) *Propionibacterium acnes* (2) *Corynebacterium* sp. (1) Other Gram-negative rods (2)

BCNE: blood culture negative endocarditis; CRT-D: cardiac resynchronization therapy–defibrillator; CRT-P: cardiac resynchronization therapy–pacemaker; DRI: device-related infection; ICD: implantable cardioverter-defibrillator; mcfDNA: microbial cell-free DNA; NV: native valve; PM: pacemaker; PV: prosthetic valve.

**Table 2 antibiotics-11-01798-t002:** Comparison of sensitivity, specificity, positive and negative predictive values between metagenomic next generation sequencing and conventional diagnostic tools.

Authors	Sensitivity	Specificity	PPV	NPV
**Cheng J et al. [14]**	mNGS: 97.6% BC: 46.2% VC: 17.1% GS: 51.4%	mNGS: 85.7% BC: 100% VC: 100% GS: 100%	mNGS: 97.6% BC: 100% VC: 100% GS: 100%	mNGS: 85.7% BC: 12.5% VC: 17.1% GS: 26.1%
**Cai S et al. [15]**	valve mNGS in culture positive and negative IE: 100%	mNGS is culture-positive IE: 100%		
**Zeng X et al. [16]**	mNGS: 85.9% BC: 29.3% VC: 16.2% Combined: 89.9%	mNGS: 72.7% BC: 100% VC: 100% Combined: 72.7%	mNGS: 96.6% BC: 100% VC: 100% Combined: 96.7%	mNGS: 36.4% BC: 13.6% VC: 11.7% Combined: 44.4%
**Santibáñez P et al. [17]**	mNGS: 88.9%	mNGS: 91.7%		
**Flurin L et al. [18]**	tMGS positive in - WB: 59% (20/34) - Plasma: 47% (16/34) - Combined: 66% (23/35) In BCPE: tMGS positive in - WB: 61% (17/28) - Plasma: 45% (13/29) - combined: 62% (18/29) In BCNE: tMGS positive in - WB: 50% (3/6) - Plasma: 45% (3/5) - Combined: 83% (5/6) BC: 83% (29) VC: 50% (6/12) 16S rRNA gene PCR on valve tissue: 60% (3/5)	Of the positive tMGS cases: 55.6% (10/18) concordant results from plasma and WB		
**To RK et al. [19]**	mNGS: 80% BC, VC and 16S rRNA: 50%	In BCPE: 100% In BCNE: 71.4%		
**Eichenberger EM et al. [20]**	mcfDNA: 87% BC: 87%			
**Olsen T et al. [21]**	NGS analysis of generators: 18% (10/57) and leads: 48% (27/56)		NGS analysis of generators (~50%) and leads (~90%)	

BC: Blood culture; BCNE: blood culture negative endocarditis; BCPE: blood culture positive endocarditis; GS: Gram staining; mNGS: metagenomic next generation sequencing; tMGS: targeted metagenomic sequencing; VC: valve culture; WB: whole blood; 16S rRNA: 16S ribosomal RNA.

**Table 3 antibiotics-11-01798-t003:** Advantages and disadvantages of different microbiologic diagnostic tools in the workup of infective endocarditis.

Microbiologic Diagnostic Tools	Advantages	Disadvantages
**Blood culture**	- Gold standard - Recovers a pathogen for identification and antimicrobial susceptibility testing - Globally available - Relatively inexpensive	- Risk of recovery of skin flora contamination - Results may be delayed (hours to days) - Limited sensitivity if prior antibiotic use (the predominate case of BCNE) - Limited sensitivity if fastidious microorganism
**Valve Culture**	- Allows identification of the pathogen and antimicrobial susceptibility testing - Widely available in laboratories	- Specimen available in a minority of patients - Time-consuming - Limited sensitivity if prior antibiotic use - Limited sensitivity if fastidious microorganisms - Limited specificity as intraoperative and/or laboratory contamination is possible
**Serology**	- Available to identify *Coxiella* spp., *Bartonella* spp., *Brucella* spp., *Legionella* spp., and fungi (*Aspergillus* spp., Histoplasma spp.) - Culture-independent - Generally available	- Regional epidemiology can impact serologic results - Sensitivity and specificity are not optimal - Titers should be correlated with clinical presentation; some serologies do not have defined cut-offs
**Polymerase chain reaction (PCR)**	- Valuable for BCNE, fastidious, difficult-to-culture, or slowly growing microorganisms - Culture-independent - Can be used on blood and valve tissue - May help clarify questionable cases with positive blood cultures that could be due to contamination - Rapid results - Large sequences available in public databases	- Does not differentiate between viable or dead bacteria, thus needs clinical correlation - Not suitable for assessment of treatment success - Contaminating free bacterial DNA - False-positive results - Lower sensitivity of broad-range PCR compared to specific PCR assays
**Fluorescence in situ hybridization (FISH)**	- Culture-independent - Allows for rapid detection and visualization of the spatial arrangement of complex microbial communities in valve tissue, and the analysis of microbial biofilms - Potential to discriminate between causative infectious agents and contaminations - Inexpensive - Rapid results after growth detection in the automated blood culture system - Has been combined with 16S rRNA-gene PCR and sequencing	- FISH assays techniques lack standardization - No antibiotic susceptibility testing - Depends on sampling accuracy - Limited availability
**Immunohistochemistry**	- Permits direct detection of microorganisms in heart valve specimens	- Does not provide identification of the genus involved - Limited availability
**16S rRNA sanger sequencing**	- Culture-independent - Determines a sample’s sequence one section at a time - Detects bacterial rRNA regardless of viability, which is advantageous for cases of prior antibiotic use	- Does not differentiate between viable or dead bacteria, thus needs clinical correlation - Identifies bacteria only
**Metagenomic next-generation sequencing**	- Culture-independent - High-throughput technique that uses parallel and simultaneous sequencing of a multitude of gene fragments to determine the sequence of nucleic acid content from a sample - Detects bacterial DNA regardless of viability, which is advantageous in cases with prior antibiotic use - Identify various microorganisms, including viruses, bacteria, fungi, and parasites - May lead to the discovery of previously undefined pathogens - Efficient and rapid sequencing approach - Potential to detect antimicrobial resistance genes - Potential to assess the microbiologic load, and the response to antimicrobial therapy - Potentially provides quicker results with earlier institution of directed antimicrobial therapy with improved outcomes	- Limited availability in clinical laboratories; delays in results if specimen send-out to a reference laboratory - Does not differentiate between viable or dead bacteria, thus needs clinical correlation - Contamination risk, dealing with background noise, bacterial identification accuracy, and subsequent clinical-biological interpretation - Expensive - Complexity in interpretation - Lack of method standardization

BCNE: blood culture negative endocarditis; BPNE: blood culture positive endocarditis; 16S rRNA: 16S ribosomal RNA; 18S rRNA: 18S ribosomal RNA.

## Data Availability

Not applicable.

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
