# Peer review of "Utility of Metagenomic Next-Generation Sequencing in Infective Endocarditis: A Systematic Review"

_antibiotics, 2022, doi:10.3390/antibiotics11121798_

Round 1

Reviewer 1 Report

The authors performed a systematic review regarding the role of mNGS in pathogen identification in IE, a subject of great interest since Infective endocarditis (IE) is a life-threatening microbial infection with an increasing prevalence and with a mortality that has not significantly improved despite technological advances.

I have the following comments:

1. First of all, I would like to thank you for your nice and interesting work

2. It is obvious that mNGS is a faster and more accurate method with certain advantages over the traditional laboratory microbial detection methods but mNGS has also limitations. To present more concisely and clearly these facts, I suggest to include a graphical representation of advantages and limitations of the method, especially since the graphical part of the manuscript is missing.

3. From clinical point of view, since limitations exist for all identification tools, currently a multimodal diagnostic approach for pathogen identification could be suggested. Based on actual knowledge regarding the method described, can you suggest an approach (algorithm) to pathogen diagnosis in IE.

4. All abbreviations used must be defined under the tables

Author Response

Reviewer 1:

  1. First of all, I would like to thank you for your nice and interesting work

Thank you for reviewing our work.

  1. It is obvious that mNGS is a faster and more accurate method with certain advantages over the traditional laboratory microbial detection methods but mNGS has also limitations. To present more concisely and clearly these facts, I suggest to include a graphical representation of advantages and limitations of the method, especially since the graphical part of the manuscript is missing.

In response to your request, a table comparing the advantages and disadvantages of currently available microbiologic diagnostic tools was included; see Table 3.

  1. From clinical point of view, since limitations exist for all identification tools, currently a multimodal diagnostic approach for pathogen identification could be suggested. Based on actual knowledge regarding the method described, can you suggest an approach (algorithm) to pathogen diagnosis in IE.

As suggested, an algorithm was included as Figure 2 to profile an approach to pathogen diagnosis. There is a qualifier and that is a fully structured sequence to address the variety of laboratory tools was not included. Rather, a list of options was depicted. This format was used because the availability of these diagnostic methods currently is highly variable among clinical laboratories. 

  1. All abbreviations used must be defined under the tables.

All abbreviations have been defined in each table of the revised manuscript. 

Sincerely,

Sara Haddad, MD

Postdoctoral Research Fellow

Division of Public Health, Infectious Diseases and Occupational Health

Department of Medicine

College of Medicine and Science

Mayo Clinic, Rochester, MN

Reviewer 2 Report

This paper is dedicated to the current trend in the diagnosis of infectious diseases such as infective endocarditis. The authors conducted a systematic review of well-known scientific papers in the English-speaking segment on the determination of the microflora of heart valves, intracardiac devices and blood by the metagenomic next-generation sequencing (mNGS) method. Diagnosis by culture and 16 sRNA sequencing was used for comparison as well.

A problem is the high incidence of "blood culture-negative endocarditis" (BCNE),
in contrast to culture methods for diagnosing infective endocarditis (from 40 to 71%).
Problems are also related to the limitations of the 16S rRNA methodology.
The authors analyze
d the advantages of mNGS for faster and more accurate diagnosis
of the causes of infective endocarditis in thi
s paper. The study was founded on a correct
methodological approach. The study criteria for inclusion and exclusion were also adjusted.
The results are obtained with an average syllable. The conclusions were valid.
The presented in paper evidence are of scientific and practical interest and serve the purpose of developing the main antibacterial therapy for difficult-to-diagnose and pharmacological treatment of infectious diseases, such as infectious endocaritis. Research in the field of studying the etiological causes of infective endocarditis and a large number of scientific groups around the world dealing with this problem, the text proposed for publication is of high relevance. The material presented in the tables is clear and concise. Results were limited to English language. Publications devoted to modern methods of diagnosing infective endocarditis are scattered both in English and in other segments of scientific publications to date. The style of presentation is understandable both for researchers and employees of the laboratory service, and for doctors of clinical medicine, which will allow a wide range of specialists to get acquainted with this study. Publications devoted to modern methods of diagnosing infective endocarditis are scattered both in English and in other segments of scientific publications to date. The style of presentation is understandable both for researchers and employees of the laboratory service, and for doctors of clinical medicine, which will allow a wide range of specialists to get
acquainted with this study.
Although mNGS diagnosis is currently not publicly available, the development of an
evidence base for the effectiveness of this method of detecting an etiological agent and
advances in molecular techniques may introduce this method as a routine method in the
near future to improve outcomes among patients with BCNE.
This systematic review, summarizing the available data in the English-language segment
of scientific publications, revealed the problematic aspects of all existing methods for
diagnosing microbial agents of infective endocarditis, which allows researchers to focus
on further development in this direction. This is especially important from the point of
view of preventing secondary and repeated episodes of infective endocarditis and
counteracting the development of antibiotic resistance in microorganisms.

Author Response

Thank you for the positive feedback. We agree that our work is “cutting edge” and focuses on a diagnostic technique that will continue to expand in its availability globally.

One additional point: several minor grammatical changes were included in the Discussion section and can be seen in the “track changes” version of the submitted manuscript.

With our detailed address of all concerns identified by the reviewers, we now hope that our revised manuscript will be accepted for publication in “Antibiotics” and thank you for the opportunity to contribute to this worthy journal.

Sincerely,

Sara Haddad, MD

Postdoctoral Research Fellow

Division of Public Health, Infectious Diseases and Occupational Health

Department of Medicine

College of Medicine and Science

Mayo Clinic, Rochester, MN